

# Association between periodontitis and COVID-19 infection: a two-sample Mendelian randomization study

Zhaoqiang Meng[1,*], Yujia Ma[2,*], Wenjing Li[1,3,4] and Xuliang Deng[1,4]

[1] Beijing Laboratory of Biomedical Materials, Department of Geriatric Dentistry, Peking University School and Hospital of Stomatology, Beijing, China
[2] Department of Epidemiology and Biostatistics, School of Public Health, Peking University, Beijing, P. R. China
[3] Peking University Health Science Center, Institute of Medical Technology, Beijing, P. R. China
[4] Key Laboratory of Dental Material, National Medical Products Administration, Beijing, China
[*] These authors contributed equally to this work.

## ABSTRACT

**Background and Objective**. Epidemiological studies report associations between coronavirus disease 2019 (COVID-19) and periodontitis; however, causality has not been proven. The aim of this study is to assess the associations between COVID-19 susceptibility and periodontitis with two-sample Mendelian randomization (MR) analyses.

**Methods**. A two-sample summary MR analysis was performed using data for outcome and exposure from the OpenGWAS database on people of European descent. Periodontal complex traits (PCTs) were chosen as a proxy for the periodontitis phenotype. The causal association between PCT3 (*Aggregatibacter actinomycetemcomitans*), PCT5 (*Porphyromonas gingivalis*), and gingival crevicular fluid (GCF) interleukin-1$\beta$ (IL-1$\beta$) and COVID-19 were considered. Genome-wide association study (GWAS) data with the two largest sample sizes were selected as COVID-19 outcomes (datasets ebi-a-GCST010776 and ebi-a-GCST010777). Single-nucleotide polymorphisms (SNPs) associated with PCT3, PCT5, and GCF IL-1$\beta$ at statistical significance at genome-wide level ($P < 5 \times 10^{-8}$) were identified as genetic instruments. We used two-sample summary MR methods and tested the existence of a pleiotropic effect with MR-Egger.

**Results**. Inverse-variance weighted (IVW) estimates showed that there was a positive association between COVID-19 risk and periodontitis (ebi-a-GCST010776: odds ratio [OR] = 1.02 (95% confidence interval (CI), 1.00–1.05), $P = 0.0171$; ebi-a-GCST010777: OR = 1.03 (95% CI, 1.00–1.05), $P = 0.0397$). The weighted median also showed directionally similar estimates. Exploration of the causal associations between other PCTs and COVID-19 identified a slight effect of local inflammatory response (GCF IL-1$\beta$) on COVID-19 risk across the two datasets (ebi-a-GCST010776: IVW OR = 1.02 (95% CI, [1.01–1.03]), $P < 0.001$; ebi-a-GCST010777: IVW OR = 1.03 (95% CI, [1.02–1.04]), $P < 0.001$). The intercepts of MR-Egger yielded no proof for significant directional pleiotropy for either dataset (ebi-a-GCST010776: $P = 0.7660$; ebi-a-GCST010777: $P = 0.6017$).

**Conclusions**. The findings suggests that periodontitis and the higher GCF IL-1$\beta$ levels is causally related to increase susceptibility of COVID-19. However, given the limitations of our study, the well-designed randomized controlled trials are needed to confirm its

Corresponding authors
Wenjing Li, wenjinglibest@sina.com
Xuliang Deng, kqdengxu-liang@bjmu.edu.cn

findings, which may represent a new non-pharmaceutical intervention for preventing COVID-19.

# INTRODUCTION

The coronavirus disease 2019 (COVID-19) worldwide pandemic stemming from outbreaks of severe acute respiratory syndrome coronavirus type 2 (SARS-CoV-2) infection triggers grim conditions such as pneumonia and respiratory failure and has been related with many deaths (*Group et al., 2021*; *Guan et al., 2020*). By November 2, 2021, >240 million cases had been confirmed, and total deaths have numbered >5.01 million globally. Given the high incidence and severity, there is an urgent need to identify risk factors related with higher susceptibility to COVID-19.

One of the most prevalent chronic inflammatory illnesses, periodontitis has a prevalence of 20–50%. Severe periodontitis is the sixth most prevalent illness worldwide (*Collaborators, 2018*). It plays a key function in stimulating inflammatory cytokine release (*Cardoso, Reis & Manzanares-Cespedes, 2018*). Therefore, if causal, periodontitis can increase the global burden of COVID-19, and interventions targeting oral inflammation will play an important role in preventing COVID-19 and its complications. Recent observational studies have found that periodontitis and its severity, gingival bleeding, and plaque accumulation (*Anand et al., 2021*; *Gupta et al., 2021b*; *Marouf et al., 2021*; *Vieira, 2021*) might be related with higher risk for severe COVID-19 (*i.e.,* hospitalization, mechanical ventilation, intubation, mortality). Moreover, the periodontium inflammation level has been related to SARS-CoV-2 infection (*Andrade et al., 2021*; *Marouf et al., 2021*). In addition, metagenomic analyses of COVID-19 patients have repeatedly revealed uncharacteristically elevated bacterial reads of *Prevotella intermedia*, a periodontal pathogen (*Zhu et al., 2020*). However, a recent study of 13,253 UK Biobank participants who had undergone a COVID-19 test yielded inadequate proof to connect periodontal disease with greater COVID-19 risk (*Larvin et al., 2020*): participants with painful gums, gums that bled, and loose teeth did not show increased COVID-19 infection, admission to hospital, or mortality risk compared to the control group. Given the global outbreak of COVID-19 and its hazards, determining whether periodontitis causes COVID-19 and targeting prevention strategies are imperative.

Mendelian randomization (MR) is the method for causal inferences in non-experimental situations with the advantage of overcoming the limitations of both observational and randomized controlled trials (RCTs) studies. It uses germline genetic variants as instrumental variable (IV) to proxy for environmentally modifiable exposures within observational epidemiological studies and is considered as analogous to RCT due to the fact that the alleles of a genetic variant are distributed randomly at meiosis (Mendel's second law of independent assortment) and individuals are then "randomized" by

nature (*Jansen, Lieb & Schunkert, 2016*; *Lawlor et al., 2008*). Observational studies may be subject to confounding factors and reverse causality, which poses challenges to causal inference. Many confounding factors might exist in these studies because periodontitis is a comorbidity with many other systemic diseases (*Hajishengallis & Chavakis, 2021*), which have been related to COVID-19 outcomes (*Ramos-Casals, Brito-Zeron & Mariette, 2021*). Therefore, it is difficult to exclude the interference of confounding factors in these studies. Further, a reverse causal association is also possible due to the coexistence of excessive inflammatory states (*Hajishengallis, 2015*; *Mangalmurti & Hunter, 2020*). Mendelian randomization (MR) can overcome confounding bias because genetic alleles are randomized to individuals at conception, thereby breaking the link with most confounders. Moreover, as genetic alleles are always assigned before disease onset, reverse causality does not affect them. And although RCT had a pivotal role in establishing causal associations, a rigorously designed, well-executed RCT is sometimes too time-consuming and expensive to be unrealistic, which makes MR design more prominent and practical especially when Genome-Wide Association Study (GWAS) summary data are publicly available. Therefore, together with vigorous observational analyses, MR analyses can yield corresponding insights to improve the assessment of possibly causal associations. Determining causal relations between COVID-19 and periodontitis might facilitate the alleviation of their influence on disease risk.

Considering the recent evidence that periodontitis patients may be more susceptible to SARS-CoV-2 infection due to a higher inflammatory burden and the infection status of periodontal pathogens, we hypothesized that periodontitis and its related phenotypes, including periodontal pathogens and gingival inflammation, are causally associated with COVID-19 susceptibility. Therefore, a two-sample MR analysis was performed to assess the causal relationship of periodontitis and its associated phenotypes with COVID-19 using OpenGWAS data (https://gwas.mrcieu.ac.uk/).

# MATERIALS & METHODS

This study is reported as per the Strengthening the Reporting of Observational Studies in Epidemiology (STROBE) guideline. No additional ethics approval was required for this study because the data are public available de-identified summary-level data. The Mendelian randomization unbiased causal associations and assumptions of this study were showed in Fig. 1. The study pipeline is illustrated on Fig. 2.

## Data acquisition for the genetic instrumental variables and study outcome

Both the genetic instrumental variables for exposure and genome-wide association study (GWAS) summary statistics for outcome were acquired from OpenGWAS, developed by the MRC IEU OpenGWAS project, the contributor of TwoSampleMR (https://github.com/mrcieu/TwoSampleMR) package and MR-base (*Hemani et al., 2018*). The data setup of the open-access OpenGWAS database is scalable, open-source, high-performance, and cloud-based, importing and publishing complete GWAS metadata and summary datasets for scientific society. The import pipeline matches these datasets to the

## Mendelian Randomization (MR)

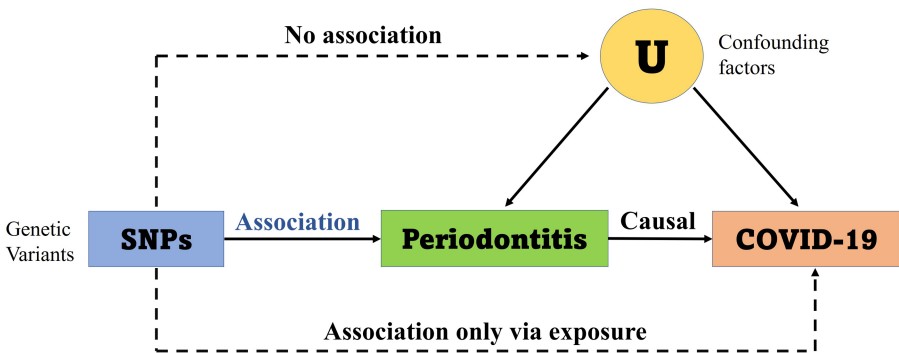

**Figure 1** **Mendelian randomization unbiased causal associations and assumptions.** Mendelian randomization (MR) is an application of the instrumental variable using genetic variants Genome-Wide Association Studies (GWAS) and Single Nucleotide Polymorphisms (SNPs). MR requires several stringent assumptions to be fulfilled. First, no confounders are associated with the genetic instrument; and second, the genetic proxy of exposure (SNP) should not be independently associated with the disease outcome but only mediates its effect *via* the relevant exposure. The SNPs used in the study had no association with the confounding variables (U) and no independent association with COVID-19.

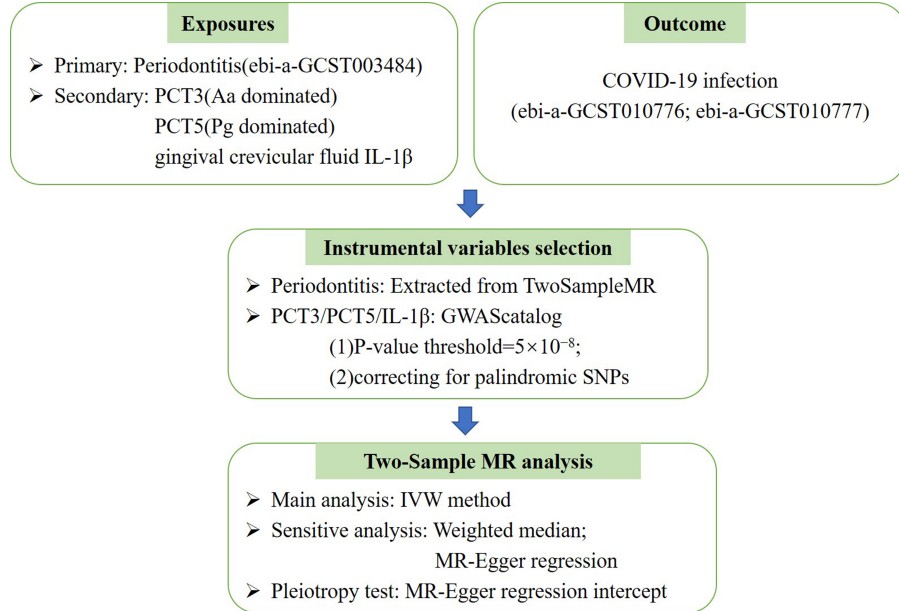

**Figure 2** **The Mendelian randomization (MR) analysis pipeline of the current study.** PCT3, periodontal complex trait dominated by *Aggregatibacter actinomycetemcomitans*; PCT5, periodontal complex trait dominated by*Porphyromonas gingivalis*; COVID-19, coronavirus disease 2019; Aa, *Actinobacillus actinomycetemcomitans*; Pg, *Porphyromonas gingivalis*; IVW, Inverse variance weighted.

reference sequence of the human genome and dbSNP, produces summary reports, and systematizes the results and metadata formats.

To decrease possible bias from stratification of the population, MR analyses were limited to people of European descent. Therefore, dataset ebi-a-GCST003484 was selected as the result of searching for "periodontitis". The dataset is derived from the study by *Offenbacher et al. (2016)*, which identified periodontal complex trait (PCT)-associated loci among 975 European-American adults participating in the Dental ARIC study. A uniformly high pathogen load, characterized as PCT1, was chosen as a proxy for the periodontitis phenotype. The outcome summary statistics were contributed by the COVID-19 Host Genetics Initiative (*The COVID-19 Host Genetics Initiative, 2020*). To improve the statistical efficiency as much as possible, GWAS datasets were selected with the two largest sample sizes as COVID outcomes (*i.e.,* ebi-a-GCST010776 and ebi-a-GCST010777). We extracted the instrumental genetic variables and their effects on the outcome by the inherent function provided in the R TwoSampleMR package, using the default parameters. The exposure–outcome datasets were matched to guarantee an identical number of single-nucleotide polymorphisms (SNPs) in the outcome and exposure sets, comparable strand alignment, accurate effect size direction, and palindromic SNP correction.

## The data source for other PCTs

In the original literature for the periodontitis phenotype, *Offenbacher et al. (2016)* refined the chronic periodontitis by enhancing clinical data with the local inflammatory response (gingival crevicular fluid interleukin-1$\beta$ [GCF IL-1$\beta$]) and microbial burden biological intermediates (*i.e.,* periodontal pathogen levels, $n = 8$), and deriving PCTs through principal component analysis. Among the six PCTs with distinctive microbial community/IL-1$\beta$ assemblies, *Aggregatibacter actinomycetemcomitans* (*Aa*) and *Porphyromonas gingivalis* (*Pg*), the two major periodontal pathogens, predominated in PCT3 and PCT5, respectively. Therefore, in addition to PCT1, we also considered the causal association between PCT3, PCT5, and local IL-1$\beta$ and COVID-19. The summary statistics were obtained from GWAS Catalog (https://www.ebi.ac.uk/gwas/home) (*Buniello et al., 2019*). The SNPs related with the corresponding phenotypes with statistical significance at the genome-wide level ($P < 5 \times 10^{-8}$) were identified as genetic instruments.

## Statistical analysis

The acceptable instrumental variables were outlined *via* three main assumptions: they associate with the relevant risk factor (relevance assumption); they and the outcome have no common cause (independence assumption); the outcome is not affected by them except *via* the risk factor (exclusion restriction assumption) (*Davies, Holmes & Smith, 2018*). Then the widely accepted inverse-variance weighted (IVW) method were used for the main analysis to estimate the causal effect between periodontitis and COVID-19. The IVW estimate is calculated by regressing the coefficient from an outcome regression on the genetic variant on that from an exposure regression on the variant, and weighting each estimate by the inverse variance of the association between the instrument and the outcome (*Bowden, Davey Smith & Burgess, 2015*). The median weighted estimates (MME)

**Table 1  List of genetic instruments for periodontitis selected by TwoSampleMR.**

| SNP | CHR | POS | Effect allele | REF allele | MAF | Overlapped gene | Beta for exposure | SE for exposure | *P-value* for exposure |
|---|---|---|---|---|---|---|---|---|---|
| rs1633266 | 1 | 159005977 | T | C | 0.32 | *IFI16* | −0.93 | 0.17 | $3.09 \times 10^{-8}$ |
| rs17718700 | 3 | 29055145 | C | T | 0.04 | – | 1.22 | 0.22 | $4.58 \times 10^{-8}$ |
| rs17184007 | 12 | 67594578 | C | T | 0.05 | – | 1.35 | 0.23 | $6.86 \times 10^{-9}$ |
| rs9557237 | 13 | 100110910 | C | G | 0.18 | – | 1.33 | 0.24 | $1.42 \times 10^{-8}$ |
| rs3811273 | 14 | 22715158 | G | A | 0.16 | – | 1.22 | 0.2 | $2.06 \times 10^{-9}$ |
| rs1156327 | 16 | 19348524 | C | T | 0.21 | – | −1.45 | 0.23 | $3.01 \times 10^{-10}$ |

**Notes.**

SNP, the label of single-nucleotide polymorphism; MR, Mendelian randomization; CHR, chromosome; POS, position; REF allele, reference allele; MAF, Minor Allele Frequency; SE, Standard Error; *IFI16, Interferon Gamma Inducible Protein 16*.

and MR-Egger regression results were also included for sensitivity analyses for the results' stability and reliability. Both methods have looser assumptions than the IVW method. MME relax the relevance assumption and only require that the weight of effective SNPs in the analysis exceeds 50% in providing valid and consistent effect estimates (*Burgess et al., 2017*); MR-Egger regression can be modified to check for pleiotropy-origin bias, and the slope coefficient obtained yields an approximation of the causal effect (*Bowden, Davey Smith & Burgess, 2015*). As it estimates an extra parameter, MR-Egger is statistically less powerful than IVW (*Choi et al., 2020*; *Zhu et al., 2016*). The existence of the pleiotropic effect were also tested using MR-Egger. An intercept significantly dissimilar from the null signifies an approximation of the effect of the genetic variants' horizontal pleiotropy (*Burgess et al., 2017*).

All MR analyses were performed using the TwoSampleMR package in R 3.6.1 (http://www.R-project.org).

## RESULTS

### Genetic instrumental variables for periodontitis

Six variants were automatically selected as primary instruments for the periodontal disease-related phenotype (GWAS ID: ebi-a-GCST003484). Table 1 presents the relationships between every genetic instrument and periodontitis. Of the six genetic instruments, rs1633266 and rs1156327 had a readily distinguishable link, having been related with autoimmune diseases across multiple ethnic populations (*Chu et al., 2011*; *Cooper et al., 2008*; *Plagnol et al., 2011*; *Zhernakova et al., 2011*). In MR analysis, rs9557237 was removed for being palindromic with intermediate allele frequencies.

### Mendelian randomization analyses for COVID-19

The causal relations between COVID-19 and periodontitis were estimated with IVW, weighted median, and MR-Egger regression methods (Table 2; Fig. 3). The two GWAS datasets for COVID-19 present the approximate results. IVW yielded proof of causal relationships between COVID-19 and periodontitis. Periodontitis was associated positively with COVID-19 risk (ebi-a-GCST010776: IVW odds ratio [OR] = 1.02 [95% confidence interval (CI), 1.00–1.05], $P = 0.0171$; ebi-a-GCST010777: IVW OR = 1.03 (95% CI

**Table 2 Mendelian randomization for periodontitis on COVID-19.**

| Methods | COVID-19 (hospitalized vs population) ebi-a-GCST010777 | | | COVID-19 ebi-a-GCST010776 | | |
|---|---|---|---|---|---|---|
| | N SNPs | OR Estimates (95%CI) | P-value | N SNPs | OR Estimates (95% CI) | P-value |
| Inverse variance weighted | 5 | 1.03(1.00–1.05) | 0.0397 | 5 | 1.02(1.00–1.05) | 0.0171 |
| Weighted median | 5 | 1.03(1.00–1.06) | 0.0295 | 5 | 1.03(1.00–1.06) | 0.0282 |
| MR Egger | 5 | 1.05(0.89–1.24) | 0.5808 | 5 | 1.06(0.93–1.21) | 0.4181 |
| MR Egger intercept | | −0.0322 | 0.7660 | | −0.0468 | 0.6017 |

Notes.
COVID, coronavirus disease 2019; ebi-a-GCST010777 and ebi-a-GCST010776, two largest COVID-19 GWAS data; SNP, the label of single-nucleotide polymorphism; $N$, sample size; OR, odds ratio; $P$, strength of evidence against the null hypothesis of no association between variant and outcome.

[1.00–1.05]), $P = 0.0397$). The weighted median also revealed estimates that were directionally similar, while MR-Egger regression appeared insufficiently powered for exceeding standard thresholds of significance (ebi-a-GCST010776: weighted median OR = 1.03 (95% CI [1.00–1.06]), $P = 0.0282$; ebi-a-GCST010777: weighted median OR = 1.03 (95% CI [1.00–1.06]), $P = 0.0295$). The intercepts of MR-Egger indicated no proof for significant directional pleiotropy for both datasets (ebi-a-GCST010776: $P = 0.7660$; ebi-a-GCST010777: $P = 0.6017$; Table 2). Our findings indicate that there are no effects of directional pleiotropy between COVID-19 and periodontitis.

We explored the causal associations between PCTs derived by principal component analysis and COVID-19 (Fig. 4). COVID-19 risk across the two datasets was slightly affected by the local inflammatory response (GCF IL-1$\beta$) (ebi-a-GCST010776: IVW OR = 1.02 (95% CI [1.01–1.03]), $P < 0.001$; ebi-a-GCST010777: IVW OR = 1.03 (95% CI [1.02–1.04]), $P < 0.001$), whereas PCT3 and PCT5 showed no statistically significant association with COVID-19.

## DISCUSSION

In the present study, the two-sample MR IVW method revealed a causal association between genetically predicted periodontitis and SARS-CoV-2 infection; that is, periodontitis increases COVID-19 susceptibility. We explored whether there is a causal association between the periodontitis-related phenotypes and COVID-19, and found that increased GCF IL-1$\beta$ levels also increase COVID-19 susceptibility. The causal role of periodontitis in COVID-19 was verified in the two largest COVID-19 GWAS datasets.

Here, we present evidence that periodontitis and GCF IL-1$\beta$ are COVID-19 risk factors. A newly published study has also explored the causal relationship between periodontitis and COVID-19 based on MR methods and reached the same conclusion that there is a causal association between periodontitis and COVID-19 (Wang et al., 2021). However, in this study, we further explored the causal effects of periodontitis-related phenotypes, including PCT3, PCT5 and inflammatory indicators IL-1$\beta$ on COVID-19, and obtained the causal association of IL-1$\beta$ in gingival crevicular fluid on COVID-19. In addition, previous epidemiological studies support our findings (Anand et al., 2021; Gupta et al., 2021b; Marouf et al., 2021; Vieira, 2021). It suggests that periodontitis and a high inflammatory burden are associated with higher COVID-19 infection, hospitalization rates and poorer

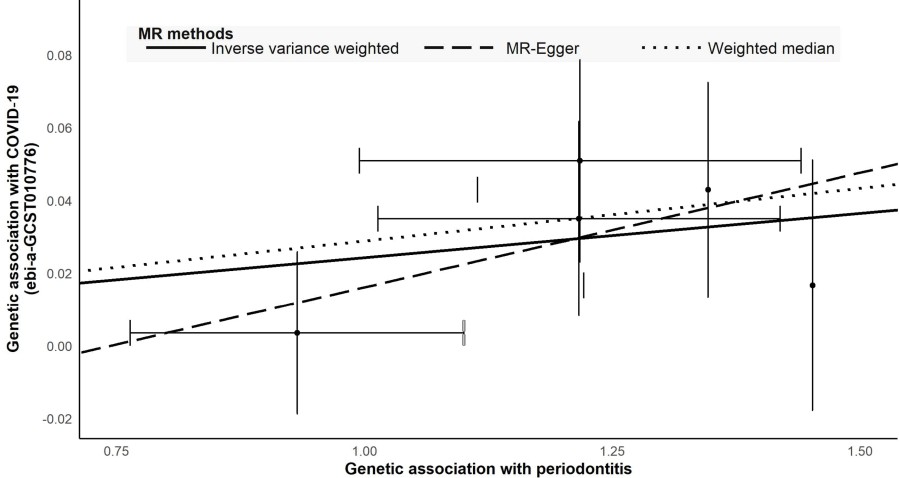

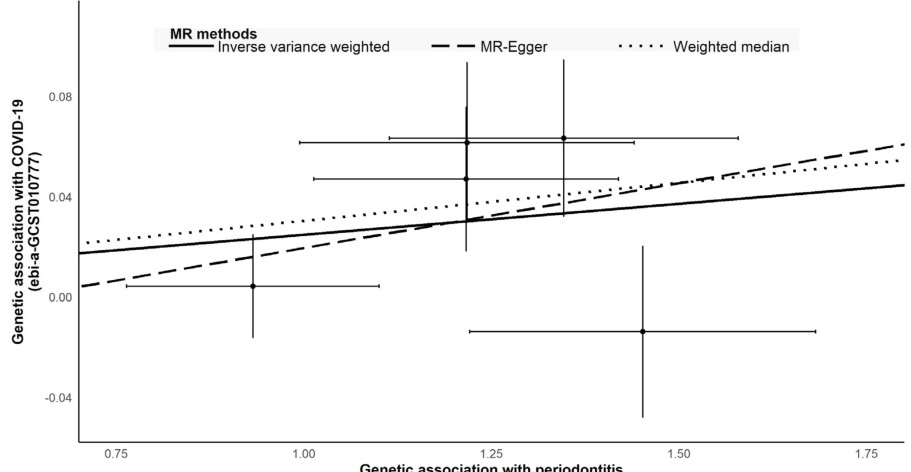

**Figure 3** **Scatter plot to visualize causal effect of periodontitis and COVID-19.** (A) COVID-19 (hospitalized *vs* population, ebi-a-GCST010777) (B) COVID-19 (ebi-a-GCST010776). The slope of the straight line indicates the magnitude of the causal association.

prognosis, which our findings support. However, the results from a large-sample study are inconsistent with our findings (*Larvin et al., 2020*). In that study, the authors found that gum pain or bleeding was not related with COVID-19 risk and that hospitalization or mortality risk were not increased by loose teeth. We speculate that there are two major reasons for the inconsistent results. First, the oral manifestations, such as gum pain, gum bleeding, and tooth loss, were self-reported and not diagnosed; the inaccurate exposure information might have led to inconsistent results. Second, confounding factors, which that study did not adjust well, might have influenced the true correlation between periodontitis and COVID-19.

SARS-CoV-2, which causes COVID-19, has been detected in the periodontium (*Fernandes Matuck et al., 2020*). The positive causal correlation between periodontitis and susceptibility to COVID-19 may be connected to the augmented viral invasion. It
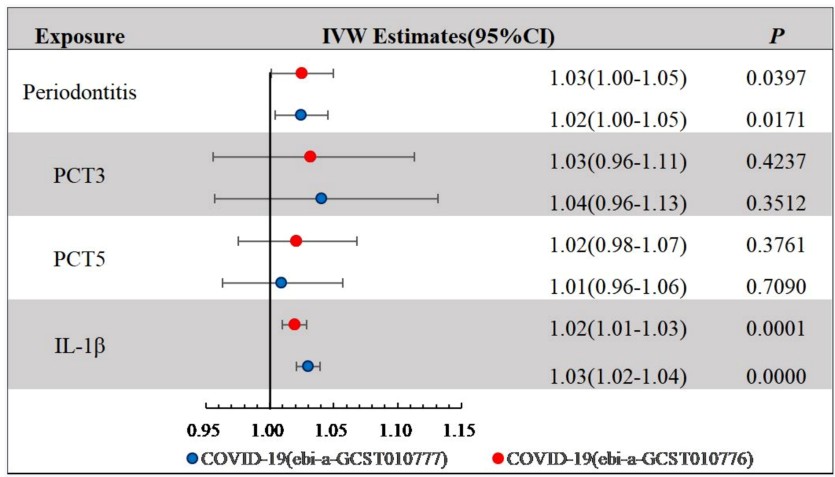

**Figure 4** **IVW Estimates for periodontal complex traits on COVID-19.** IVW, Inverse variance weighted; COVID, coronavirus disease 2019; PCT3 and PCT5 were dominated by *Aggregatibacter actinomycetem-comitans* and *Porphyromonas gingivalis*, respectively.

is thought that periodontitis-induced epithelial mucosal ulcers may cause the loss of defenses against SARS-CoV-2. This renders periodontitis patients more vulnerable to infection by SARS-CoV-2 (*Pfutzner, Lazzara & Jantz, 2020*). Recently, it was hypothesized that the oral immune barrier breakdown, which can take place in periodontitis, may result in SARS-CoV-2 dissemination into the systemic circulation through its GCF and saliva oral reservoirs (*Gupta et al., 2021a*; *Graham Lloyd-Jones et al., 2021*). In addition, this association may be associated with the dysregulated inflammatory response. Cytokine storms caused by inflammation are important drivers of COVID-19 (*Mangalmurti & Hunter, 2020*). Periodontitis, one of the most common chronic inflammatory diseases, induces a dysregulated inflammatory response (*Hajishengallis, 2015*). Subgingival plaque biofilm interacts with host cells, leading to the release of proinflammatory cytokines and aggravating systemic inflammation (*Pan, Wang & Chen, 2019*). Therefore, it might cause overstimulation of the immune system, thereby aggravating SARS-CoV-2 infection severity because of the cytokine storm. Our findings that increased GCF IL-1$\beta$ levels were associated with increased COVID-19 also support this assumption. GCF IL-1$\beta$ might be a potential biomarker for predicting COVID-19 occurrence. However, exactly how IL-1$\beta$ affects COVID-19 remains unclear, and the association between GCF IL-1$\beta$ and SARS-CoV-2 infection warrants further exploration. Some studies have proposed that molecular factors in healthy oral mucosa might enable virus-host cell interactions. Gingival fibroblasts and the periodontal ligament contain angiotensin-converting enzyme 2 (ACE2), which has an affinity for SARS-CoV-2 and penetrates target cells easily (*Madapusi Balaji et al., 2020*). Therefore, ACE2-expressing cells may be vulnerable to infection.

It is worth mentioning that we found no causal association between PCT4 (*Aa*) and PCT5 (*Pg*) with COVID-19. Chakraborty et al. found that metagenomic analysis of patients with SARS-CoV-2 infection repeatedly yielded uncharacteristically high *P. intermedia* bacterial

reads and failed to detect higher *Aa* and *Pg* reads (*Zhu et al., 2020*). This suggests that there might be no direct causal relationship between COVID-19 and these two periodontal pathogens. However, the specific correlation requires further exploration.

In the present two-sample MR study, we established a causal relationship between periodontitis and COVID-19 susceptibility, excluding the effects of confounders and reverse causality. The finding is important because we have established a link between periodontitis and COVID-19 risk from a high evidence level and broadened the understanding of COVID-19 risk. Therefore, periodontitis is a COVID-19 risk factor, which may contribute to the development of effective directives and strategies for controlling COVID-19 spread to vulnerable populations. Furthermore, we demonstrate for the first time that higher GCF IL-1$\beta$ levels elevate COVID-19 risk, suggesting a novel mechanism through which SARS-CoV-2 might cause an individual onset of COVID-19 through the IL-1$\beta$-related pathways. Therefore, interventions targeting periodontal inflammation might also act in COVID-19 prevention. In addition, we demonstrate the result's stability by using a causal relationship between periodontitis and COVID-19 in two databases.

However, our study has some limitations. The main limitation of our study is the insufficient power due to the relatively small sample size of GWAS for PD. Although the heritability of dental caries and periodontitis has been reported to be as high as 50%, GWAS with small sample sizes limits the ability to extract more significant instrumental variables, which results in the instrumental variables we used only explaining a small portion of phenotypic variation (*Shungin et al., 2019*). We partly verified the robustness of the results by applying multiple MR methods in parallel, while a larger sample size of GWAS for PD was still expected to validate the findings in our study. In addition, MR analysis relied on periodontitis exposure attributable to genetic predisposition, which is a lifetime exposure level and thus does not reflect the short-term effect of periodontitis and its components on COVID-19 risk. Therefore, the short-term exposure risk needs to be verified by further clinical studies. Third, due to limited data resources, our results are largely built on European populations and do not characterize the general inferences for other ethnic groups. While we assessed the causality of COVID-19 according to current data and numerous corresponding approaches, these results should be authenticated with added clinical resources and detailed investigation of the underlying causality mechanism.

## CONCLUSIONS

The findings suggests that periodontitis and the higher GCF IL-1$\beta$ levels is causally related to increase susceptibility of COVID-19. However, given the limitations of our study, the well-designed randomized controlled trials are needed to confirm its findings, which may represent a new non-pharmaceutical intervention for preventing COVID-19.

## ACKNOWLEDGEMENTS

We thank all the researchers for making the summary data publicly available, and we are grateful for all the investigators and participants who contributed to those studies.

### Funding

This work was supported by the National Natural Science Foundation of China (82201062) and the National Science and Technology basic resources project (2018FY101004). The funders had no role in study design, data collection and analysis, decision to publish, or preparation of the manuscript.

### Grant Disclosures

The following grant information was disclosed by the authors:
National Natural Science Foundation of China: 82201062.
National Science and technology basic resources: 2018FY101004.

### Competing Interests

The authors declare there are no competing interests.

### Author Contributions

- Zhaoqiang Meng performed the experiments, analyzed the data, prepared figures and/or tables, and approved the final draft.
- Yujia Ma performed the experiments, analyzed the data, prepared figures and/or tables, and approved the final draft.
- Wenjing Li conceived and designed the experiments, authored or reviewed drafts of the article, and approved the final draft.
- Xuliang Deng conceived and designed the experiments, authored or reviewed drafts of the article, and approved the final draft.

### Data Availability

The COVID datasets are available at the University of Bristol:

(1) COVID-19 Host Genetics Initiative, COVID-19 (RELEASE 4), Dataset: ebi-a-GCST010776, https://gwas.mrcieu.ac.uk/datasets/ebi-a-GCST010776/, DOI: 10.1038/s41431-020-0636-6, PMID: 32404885.

(2) COVID-19 Host Genetics Initiative, COVID-19 (hospitalized *vs* population) RELEASE 4, Dataset: ebi-a-GCST010777, https://gwas.mrcieu.ac.uk/datasets/ebi-a-GCST010777/, DOI: 10.1038/s41431-020-0636-6, PMID: 32404885.

The periodontitis datasets are available at the GWAS Catalog: *Offenbacher et al. (2016)*. Genome-wide association study of biologically informed periodontal complex traits offers novel insights into the genetic basis of periodontal disease. https://www.ebi.ac.uk/gwas/publications/26962152, DOI: 10.1093/hmg/ddw069,

- Periodontal disease-related phenotype: PMID: 26962152, GWAS catalog ID: GCST003484.

- PCT3: PMID: 26962152, GWAS catalog ID: GCST008440b.

- PCT5: PMID: 26962152, GWAS catalog ID: GCST008442.

The gingival crevicular fluid interleukin-1β [GCF IL-1β] data is available at *Offenbacher et al. (2016)*. GWAS for Interleukin-1β levels in gingival crevicular fluid identifies IL37 variants in periodontal inflammation. https://www.ebi.ac.uk/gwas/studies/GCST007542, DOI: 10.1038/s41467-018-05940-9, PMID: 30206230, GWAS catalog ID: GCST007542.

## Supplemental Information

Supplemental information for this article can be found online at http://dx.doi.org/10.7717/peerj.14595#supplemental-information.

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
