# Peer review of "Association between periodontitis and COVID-19 infection: a two-sample Mendelian randomization study"

_PeerJ, doi:10.7717/peerj.14595_

## Round 0.1 · original submission · Major Revisions

Dear Authors

Kindly respond to the reviewer's Queries.

Best Regards
Dr Mallineni

·

Basic reporting

General comments
This study innovatively attempts to establish the causative relationship, if any, between the coronavirus disease and periodontitis. However, some revisions are suggestible for the fine tuning of it technical presentation.
The use of possessive pronouns such as “we” and “our”, for example “134 Therefore, we performed a 2-sample MR analysis to assess the causal relationship of 135 periodontitis” is not too ethical for technical writing and should be revised for the entire manuscript. The above example can be restructured as “134 Therefore, a 2-sample MR analysis to assess the causal relationship of 135 periodontitis was performed...”

Experimental design

The manuscript is presented with sufficient originality.

Graphical abstract
A particular graphical abstract showing a summarizing and figurative expression of the whole study would be greatly beneficial. The authors should strongly consider adding this to the manuscript.

Validity of the findings

Conclusions
Only ONE conclusion was made, and as such this portion should not be termed as a conclusion. More work should be done in this regard. The conclusions should arise from the output of every variable taken into consideration and studied in the course of the experimentation. Furthermore, there should be recommendation for subsequent works related to this subject.

Reviewer 2 ·

Basic reporting

Well written, organized, introduction leads to the relevant hypothesis.

Experimental design

Acceptable.

Validity of the findings

Satisfactory.

Additional comments

It is suggested to refer the following publication, if possible, and add it in the discussion to further strengthen the context of the article.
Wang Y, Deng H, Pan Y, Jin L, Hu R, Lu Y, Deng W, Sun W, Chen C, Shen X, Huang XF. Periodontal disease increases the host susceptibility to COVID-19 and its severity: a Mendelian randomization study. J Transl Med. 2021 Dec 24;19(1):528.

Reviewer 3 ·

Basic reporting

Recent references should be added.
Language is clear.
You may find my detail comments at the additional comment section

Experimental design

method should be described more details.
You may find my detail comments at the additional comment section.

Validity of the findings

You may find my detail comments at the additional comment section.

Additional comments

Dear Author,
Thank you very much for submitting your paper with the title “The causal relationship between periodontitis and 1 COVID-19 infection: a 2-sample mendelian randomization study” to PeerJ.
You may find my comment below.
*There is a recent study (you may find below), assessed the COVID and periodontal disease by Mendelian randomization. Please refer to the mentioned paper and discuss the similarities and differences.

Wang, Y., Deng, H., Pan, Y. et al. Periodontal disease increases the host susceptibility to COVID-19 and its severity: a Mendelian randomization study. J Transl Med 19, 528 (2021). https://doi.org/10.1186/s12967-021-03198-2

*Please add some informative sentences about Mendelian randomization research method at the introduction section.
*Line 106: “Currently, observational studies yield insufficient levels of evidence, and performing randomized controlled trials (RCTs) is unrealistic.”
It is a very sharp sentence. Please make the sentences more informative and comparative, also add a reference. RCT is the gold standard approach to search for causal relationships, but well-designed RCT has some disadvantages like …..
-*Line 261 Please correct the spacing at the related sentence.

*Mendelian randomization is a method of using measured variation in genes of known function to examine the causal effect of a modifiable exposure on disease observational studies. Extensions of the MR approach offer a potentially useful method for strengthening causal inference in epigenetic, but there are some limitations.
The study design has certain challenges which can hamper the interpretation of the results. l have some important questions:

- The sample size of GWAS for PD was small.
-One of the most significant limitations is pleiotropy. Pleiotropy, where a genetic variant has more than one direct correlate that would invalidate conclusions based on the assumption of a single pathway, is an important issue in MR.
- Epigenetics are still relatively limited.
- The MR findings only reflect the change in COVID-19 risk due to a genetically predisposed status of Periodontal disease; the short-term effect of periodontitis on COVID-19 risk is unknown.
-The data are often generated on diseased tissue and that they usually have very little information available on environmental exposures or other relevant covariates.
-Mendelian randomization is that precise estimates of causal effect are often biased. For example, causal effect estimates from Mendelian randomization studies can be thought of as a population-average effect (i.e., as if the intervention was applied to the entire population) and could be different than the effect of interventions applied to specific subgroups.

The manuscript should be rewritten related to mentioned concerns above.

---

## Round 0.2 · Minor Revisions

The manuscript is almost ready to be accepted but the title of the submission claims to show a causal relationship between periodontitis and COVID-19 infection. However, the authors did not perform an experiment that would allow them to make a causal inference, so I ask that the title be amended to:

"Association between periodontitis and COVID-19 infection: a 2-sample mendelian randomization study"

Best Regards

·

Basic reporting

Manuscript has been improved for its technical soundness.

Experimental design

Manuscript is seemingly satisfactory.

Validity of the findings

Manuscript is seemingly satisfactory.

Reviewer 3 ·

Basic reporting

According to my opinion, after recent modifications, the article is ready for publication.

Experimental design

No comment

Validity of the findings

No comment

Additional comments

No Comment

---

## Round 0.3 · accepted · Accept

Congratulations Authors.
Thank you for choosing PeerJ.
Best Regards